# Flow Away your Differences:
# Conditional Normalizing Flows as an Improvement to Reweighting

Malte Algren[1][†], Tobias Golling[1], Manuel Guth[1], Chris Pollard[2] and John Andrew Raine[1][⋆]

**1** University of Geneva, Geneva, Switzerland
**2** University of Warwick, United Kingdom
[†] malte.algren@unige.ch [⋆] john.raine@unige.ch

April 28, 2023

## Abstract

We present an alternative to reweighting techniques for modifying distributions to account for a desired change in an underlying conditional distribution, as is often needed to correct for mis-modelling in a simulated sample. We employ conditional normalizing flows to learn the full conditional probability distribution from which we sample new events for conditional values drawn from the target distribution to produce the desired, altered distribution. In contrast to common reweighting techniques, this procedure is independent of binning choice and does not rely on an estimate of the density ratio between two distributions.

In several toy examples we show that normalizing flows outperform reweighting approaches to match the distribution of the target. We demonstrate that the corrected distribution closes well with the ground truth, and a statistical uncertainty on the training dataset can be ascertained with bootstrapping. In our examples, this leads to a statistical precision up to three times greater than using reweighting techniques with identical sample sizes for the source and target distributions. We also explore an application in the context of high energy particle physics.

# 1  Introduction

In many areas of science, simulations are used to depict the behaviors of real world systems. However, due to lack of knowledge or approximations applied to make the simulation tractable, predicted distributions often deviate from observations at some level of precision. For accurate tests of theoretical predictions using data collected from experiments, it is therefore necessary to correct the simulated data and adjust the distributions to better match observed data. Correcting observed mis-modelling is particularly important in high energy particle physics (HEP), in which Monte Carlo simulation (MC) of complex processes form the bedrock of tests of the Standard Model of particle physics (SM); for example see Refs. [1–3].

One way of improving a multi-dimensional density $p$ is to alter the initial marginal densities of some quantities $p(c)$ to better reflect a target $q(c)$; $q(c)$ may represent an observation or ground truth of some control variable, for example. Quantities of interest, $x$, will have simulated and true conditional densities on $c$, respectively $p(x|c)$ and $q(x|c)$. A new joint density can be constructed from the conditional density of $p$ together with the marginal density over $c$ of $q$: $p'(c,x) = p(x|c)q(c)$. Following the nomenclature of Cover and Thomas, the relative entropy between any two joint densities $f$ and $g$ is given by

$$D(f(c,x) \,||\, g(c,x)) = D(f(c) \,||\, g(c)) + D(f(x|c) \,||\, g(x|c)),$$

proven as Theorem 2.5.3 in Ref. [4]. From this is it clear that

$$D(p'(c,x) \,||\, q(c,x)) \leq D(p(c,x) \,||\, q(c,x)), \tag{1}$$

since $\forall f, g. \, D(f \,||\, g) \geq 0$, while $D(p'(c) = q(c) \,||\, q(c)) = 0$. In other words, the divergence between $p'(c,x)$ and $q(c,x)$ comes entirely from any residual difference in their conditional densities, $D(p(x|c) \,||\, q(x|c))$.

The standard approach in HEP to correct or alter a distribution is to derive the density ratio estimates (DREs) between the initial and target distributions in some space. The density ratio at a point $c$ is defined as $q(c)/p(c)$, where $q$ and $p$ are the two densities being compared. These DREs can be applied as weights to samples of the predicted density $p$ to improve the matching of the simulated distribution to the some other distribution $q$ – the observed data, for example. Alternatively, the DREs can be used to down-sample the initial distribution to match the target. Histograms or classifiers are commonly used to estimate the density ratios and derive weights for each event.

In this work we explore the use of conditional Normalizing Flows (cNFs) [5–10] to adjust the distributions of samples drawn from a simulator. We demonstrate that using cNFs, we can achieve better closure than both binned and unbinned reweighting approaches on a non-trivial toy example, and demonstrate its further possible applications to a HEP example, correcting the kinematics of top quark pairs produced at the LHC.

## 2 Method

### 2.1 Reweighting

In HEP, the standard approach to match distributions over $c$ and propagate to other distributions over quantities of interest $x$ is to derive the DREs as a function of $c$ ($R(c) = q(c)/p(c)$ for an initial distribution $p$ and a target distribution $q$). The values $R(c)$ are applied to samples drawn from the initial distribution $c, x \sim p(c, x)$ as weights, such that the relative entropy between the altered probability distribution matches $p'(c, x) = p(x|c)q(c)$. As shown in Equation 1, $p'(c, x)$ is guaranteed to have a lower relative entropy with respect to the target distribution $q(c, x)$ than is observed between the initial distribution $p(c, x)$ and $q(c, x)$. Given the target distribution $q(c, x)$ has values $c \sim q(c)$, it follows that with weights calculated as

$$R(c) = \frac{q(c)}{p(c)},$$

and starting from the definition of the joint distributions

$$p_{c \sim p(c)}(c, x) = p(x|c)p(c), \tag{2}$$

it can straightforwardly be shown that

$$p(c, x)R(c) = p(x|c)p(c)R(c),$$
$$p(c, x)R(c) = p(x|c)p(c)\frac{q(c)}{p(c)},$$
$$p(c, x)R(c) = p'(c, x).$$

In deriving the weights $R(c)$, one needs access only to the marginal distributions $p(c)$ and $q(c)$, which can for instance be calculated from the samples drawn from the two distributions.

Two methods are commonly used used to extract $R(c)$ in HEP. Perhaps the simplest is to build the ratio from two binned histograms, with the benefit that they are straightforward to calculate. Problems arise when the true $R(c)$ changes on a scale smaller than the width of the histogram bins; in large-dimensional spaces with a finite number of samples from the initial and target distributions, one is often forced to choose between small bin sizes with an imprecise DRE estimate in each or large bins over which $R(c)$ may change substantially. This may be mitigated somewhat through sequential application of weights calculated in each dimension separately, but this procedure correctly produces the target distribution only in the limit that the joint densities factorize into a product of marginals.

A second approach is to approximate $R(c)$ using neural networks. As presented in Ref. [11], it is possible to train a classifier $f_\phi(c)$, by minimizing over the classifier parameters $\phi$, to discriminate between samples drawn from the distributions $p(c)$ and $q(c)$, from which the resulting classifier output can be converted into a weight with

$$R(c) = \frac{f_\phi(c)}{1 - f_\phi(c)}. \tag{3}$$

This has advantages over the binned approach in that it is fully continuous and can correlate weights at the required length scales in order to build the distribution $p'(c, x)$.

However, both approaches based on DREs share the same drawback: they rely on building an approximate ratio of two probability distributions rather than simply approximating $p(x|c)$ and altering the density over $c$. If there is little overlap between the support of the two distributions this leads to either very large or very small values of $R(c)$, resulting in an effective loss of the sample size once reweighting is applied. This can have a large impact on the statistical precision of $q(c, x)$.

## 2.2 Conditional normalizing flows

Normalizing flows are a family of generative models which in recent years have gained traction due to their ability to map the probability distribution of a complex distribution from a simpler distribution of the same dimensionality. They learn the density of the complex distribution under a series of invertible transformations $f_\theta$, a family of functions parameterized by $\theta$, by minimizing the loss function

$$\log(p_X(x)) = \log\left(p_Z(f_\theta^{-1}(x))\right) - \log\left|\det\left(\mathcal{J}(f_\theta^{-1}(x))\right)\right|$$

with respect to $\theta$, where $\mathcal{J}\left(f_\theta^{-1}(x)\right)$ is the Jacobian of the transformation $f_\theta^{-1}(x)$, which maps the complex distribution $p_X$ to the chosen base distribution $p_Z$. Conditional normalizing flows similarly follow from the change of variables formula, with the probabilities and transformation conditional on some additional properties $c$

$$\log(p_X(x|c)) = \log\left(p_Z(f_\theta^{-1}(x|c))\right) - \log\left|\det\left(\mathcal{J}(f_\theta^{-1}(x|c))\right)\right|. \tag{4}$$

Due to their ability to learn the conditional probability distribution $p(x|c)$ from samples drawn from $p(c, x)$, we demonstrate normalizing flows can be used directly with Eq. 2 to perform the same role as reweighting. Instead of learning or approximating the density ratio, one needs only to sample from the normalizing flow but with the target distribution $q(c)$ to produce the desired distribution $q(c, x)$.

There are several advantages to this approach over constructing DREs and reweighting. The statistical precision of regions of probability space are no longer constrained by the derived DREs, as all sampled events have a weight of unity. Instead, the statistical uncertainty comes only from the statistical precision of the learned conditional density $p(x|c)$, as it is possible to sample any number of events from it for each $c$. Where necessary, the statistical uncertainty on the sampled distribution can be estimated through bootstrapping [12].

In our tests, the normalizing flows are able to learn a more precise approximation of the conditional probability distribution $p(x|c)$ at low probability values for $c$ than is possible when considering only data with those values $c$. This is a result of the normalizing flow learning the correlations in the underlying probability distribution function. Similar behaviour is observed when training neural networks to approximating the DRE in comparison to binned approaches.

## 3 Related work

Modifying distributions to account for mis-modelling or to follow some other underlying distribution is often performed with event weights or resampling.

In the case of reweighting, machine learning approaches based on density ratio estimation have had a large amount of success, in particular in the field of high energy physics. The CARL method, introduced in Ref. [11], uses neural networks to learn the likelihood ratio which can be used for covariate shift and importance sampling of distributions. In Ref. [13], boosted decision trees are employed to derive event weights for reweighting. In Ref. [14], neural networks are used to reweight the full phase space of events generated with two different MC generators, and extended to the conditional case in Ref. [15].

Instead of reweighting events, it is also possible to learn a mapping between the initial and target distributions. In Ref. [16] partially input convex neural networks [17] are used to calibrate distributions from MC simulation to match those observed in data. In Refs. [18, 19] normalizing flows are trained to transport events from MC to data domain to another, and in Ref. [20] they are used to move to data to different conditional values on the same distribution.

In our work we use normalizing flows to learn the conditional probability distribution of the initial data, from which events following a different conditional distribution can be sampled. This is similar to the approach used in Ref. [21]. Here, a normalizing flow is trained to learn the conditional density on side-band data and apply it to a blinded signal region. These data are used to train a classifier, and no treatment is required or derived to account for the statistical uncertainty on the relating distribution. Concurrent with our work, Ref. [22] presents an examination of similar methods applied for estimating backgrounds in blinded signal regions, including estimation of the statistical uncertainty.

## 4 Application to a toy example

To show the benefit of cNFs in comparison to traditional reweighting approaches, we use a toy example from which any number of data can be sampled for any chosen change of conditions. As shown in Ref. [23], sampling more data points from a generative model than are available in the initial dataset can lead to higher statistical precision in the output of trained models.

We define a two dimensional probability distribution $f(c)$ solely dependent on two conditional variables $c = (c_0, c_1)$, with marginals $f_i$, $i \in 0, 1$

$$f_0(c_0', c_1') = \frac{c_0'}{(1 + c_1'^2)} - \frac{c_1'}{(1 + c_0'^2)} \qquad \text{with} \qquad c_0', c_1' = \mathcal{N}\left(\begin{pmatrix} c_0 \\ c_1 \end{pmatrix}, \begin{pmatrix} 2 \cdot c_0, c_1 \\ c_0, c_1/2 \end{pmatrix}\right) \qquad (5)$$

$$f_1(c_0', c_1') = \frac{(c_0' + c_1')^2}{(1 + c_0' + c_1')} - \frac{(c_0' - c_1')}{(1 + c_0'^2)} \qquad \text{with} \qquad c_0', c_1' = \mathcal{N}\left(\begin{pmatrix} c_0 \\ c_1 \end{pmatrix}, \begin{pmatrix} c_0, c_1/2 \\ c_0, c_1/2 \end{pmatrix}\right) \qquad (6)$$

from which samples $x$ are drawn. Here $c_0'$ and $c_1'$ are distributions defined by $c_0$ and $c_1$.

For the initial probability distribution, we choose a 2D standard normal distribution for $p(c)$, $c \sim \mathcal{N}(0, \mathbb{1})$. Figure 1 shows the 1D marginals of $x_0$ and $x_1$ for samples drawn from $f(c)$. For target distributions we change the distribution from which $c$ is sampled. Here we use a skewed Gaussian distribution, with

$$c_1 \sim \mathcal{SN}(1.5, 1.5, -2.5) \qquad \text{and} \qquad c_2 \sim \mathcal{SN}(1.5, 1.5, 2.5),$$

where the three parameters to $\mathcal{SN}$ are the location, scale, and shape, respectively. We also test a symmetric smooth box distribution defined by

$$c_{1/2} \sim q(x) = \frac{1}{1 + \exp(-10(x + 1))} - \frac{1}{1 + \exp(-10(x - 1))}.$$

Each target distribution has a sample size of $10^7$, such that the statistical uncertainty on the target distribution does not impact the performance of the three methods.

We train a conditional normalizing flow on the initial samples, learning the conditional density $p(x|c)$ from samples $x \sim f(c)$. To produce the target distributions with different conditions, we sample once from the cNF for all $10^7$ values of $c$ from the target conditions to generate the target distribution under new conditions.

We compare the cNF approach to two reweighting references. For *DRE (binned)* we take a ratio of the initial and target distributions over $c$ in two dimensions, and for *DRE (NN)* we train a classifier following the procedure in Ref. [11] with $c$ as an input. The distributions of $f(c)$ for the three scenarios are shown in Fig. 1.

### 4.1 Configuration

We construct our conditional normalizing flows using 5 stacks of rational quadratic splines [24] with autoregressive transformations for each layer. The cNFs are implemented using the

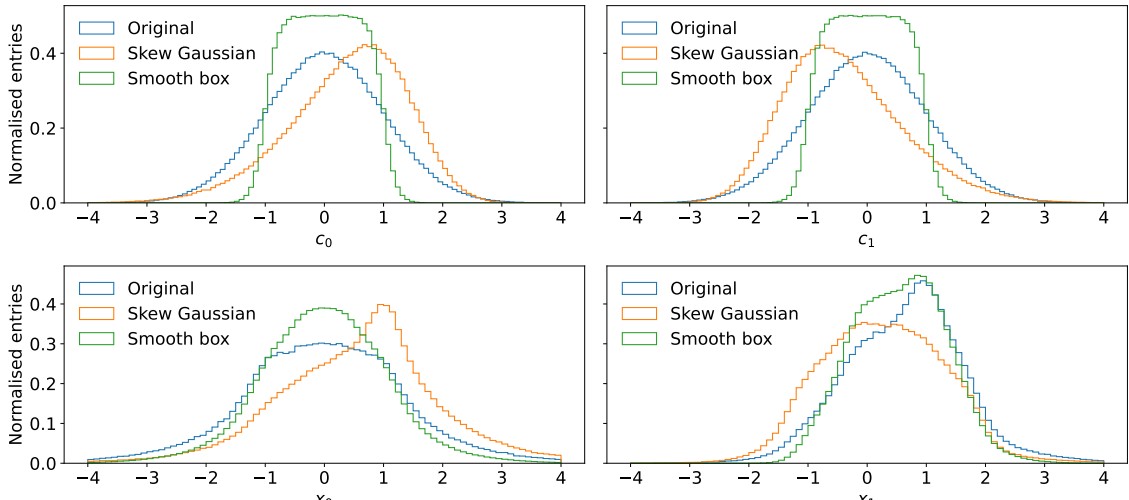

Figure 1: The ground truth distributions of the conditional quantities $c_0$ and $c_1$ (upper panels) as well as the marginals of the quantities of interest, $f_0$ and $f_1$ (lower panels), for three choices of distribution over $c_0$ and $c_1$: a standard multivariate normal Gaussian distribution, a skewed Gaussian, and a uniform distribution over $[-1, 1]$ for each choice of $c_i$.

`nflows` [25] package with PyTorch [26]. The networks are trained for 500 epochs with a cosine annealing learning rate [27], with an initial value of $10^{-4}$ using the `Adam` optimiser [28] and a batch size of 512.

For both of the reweighting approaches, event weights are calculated for each target distribution. For DRE (binned), the weights are calculated by taking the ratio of the target and initial histograms, defined using a uniform grid with 100 bins between [-4,4] on each axis in $c$. For DRE (NN), the classifier trained on $c$ from the target and initial distribution comprises four layers of 64 nodes, with `ReLU` activations in the hidden layers and a sigmoid activation on the output. It is trained with a flat learning rate of $10^{-4}$ for 50 epochs using the `Adam` optimiser. Event weights are defined using Eq. 3 with the classifier output.

## 4.2 Comparison of methods

In Fig. 2 we show the distribution of data drawn from the cNF trained on the initial data, but following the Skew Gaussian and Smooth box distributions. The statistical uncertainty is obtained with bootstrapping, trained 12 cNFs on bootstrapped training data. The central estimation is taken as the mean over all 12 models. We compare this to the closure obtained by the DRE (binned) and DRE (NN) approaches shown in Figs. 3 and 4 for the skewed Gaussian and smooth box target conditional distributions respectively. In these figures, we look at not only the closure between the reweighting method and the target, but also the relative statistical precision in each bin with respect to the cNF. The statistical uncertainty on the reweighting approaches is calculated from the sum of weights in each bin.

For both target distributions, the agreement of all approaches is reasonable, however we observe that the cNF is much closer to the true target distribution than the reweighting approaches. This is most noticeable in the tails of the distribution. In regions of the distribution with low numbers of events, we observe fluctuations in all three approaches about the true target distribution. In regions of high yields, we see that the cNF is matching the true target distribution perfectly whereas the reweighting approaches, although still within the uncertainties of the prediction, show less precise closure. Furthermore, the statistical precision when

using the cNF is almost always greater than either reweighting approaches.

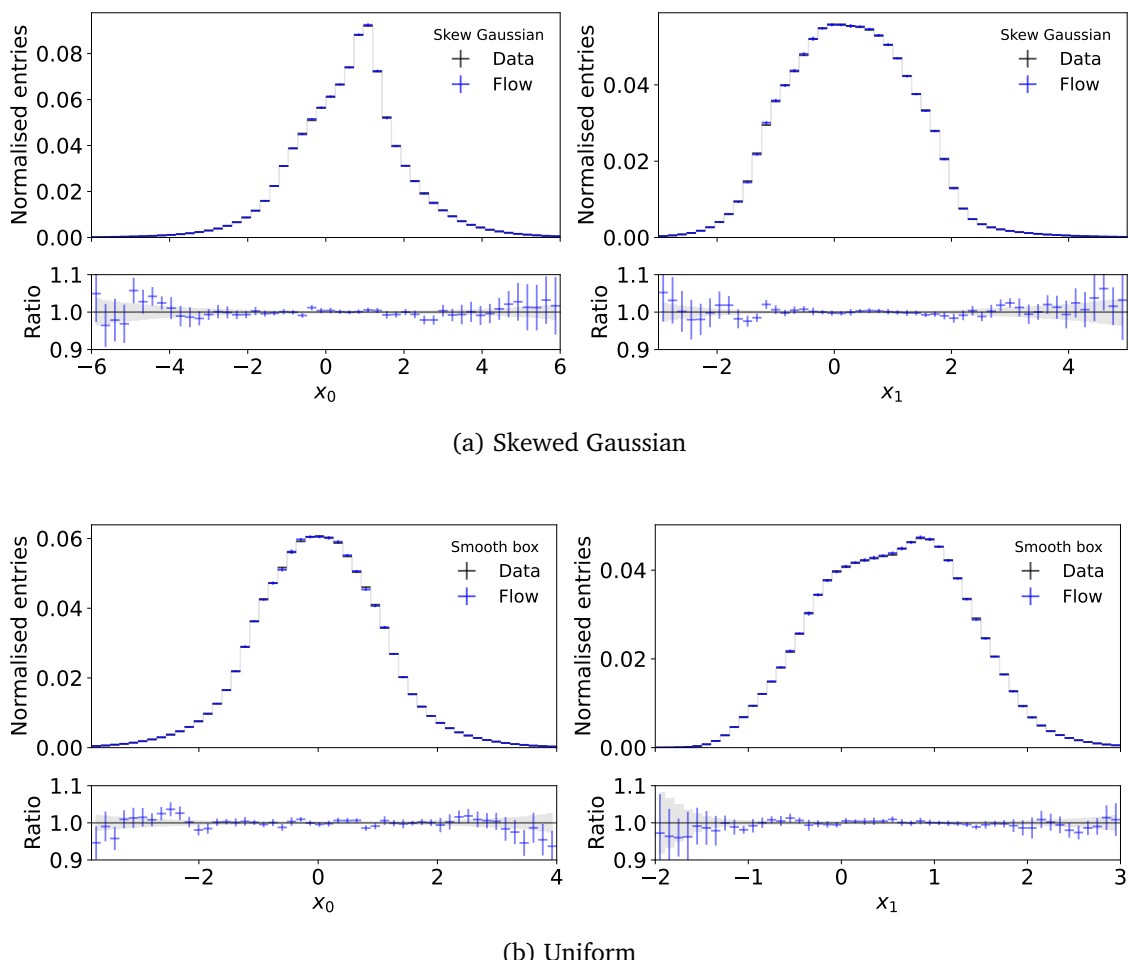

(a) Skewed Gaussian

(b) Uniform

Figure 2: The marginals of the cNF trained on the original Gaussian conditional distributions applied to a skewed Gaussian or uniform conditional distribution compared to the ground truth. Good closure is observed for both marginals. Statistical uncertainties in the cNF function are estimated using 12 bootstraps.

As a further measure of performance we train additional classifiers on the output of each of the three approaches to separate them from the target distribution. In the case where the distributions match perfectly the classifier will not be able to separate the samples, whereas the worse the closure the easier it will be. All methods result in a much better closure than the initial distribution, with area under the receiver operator characteristic curve close to 0.5 in all three cases.

As an additional test we want to verify whether the normalizing flow truly learns the conditional probability density from the training samples. In Fig. 5 we draw samples from a narrow window in $c$ following the original distribution over conditions used to train the normalizing flow. These data are compared to samples drawn from the initial distribution for the same values $c$. Here we can see that samples from the normalizing flow closely match the original target distribution within statistical uncertainties.

## 4.3 Validating statistical uncertainties

To verify that the statistical uncertainties associated with the methods are reasonable, we look at the pulls observed in all bins of the two-dimensional distributions. The pull per bin $i$ is

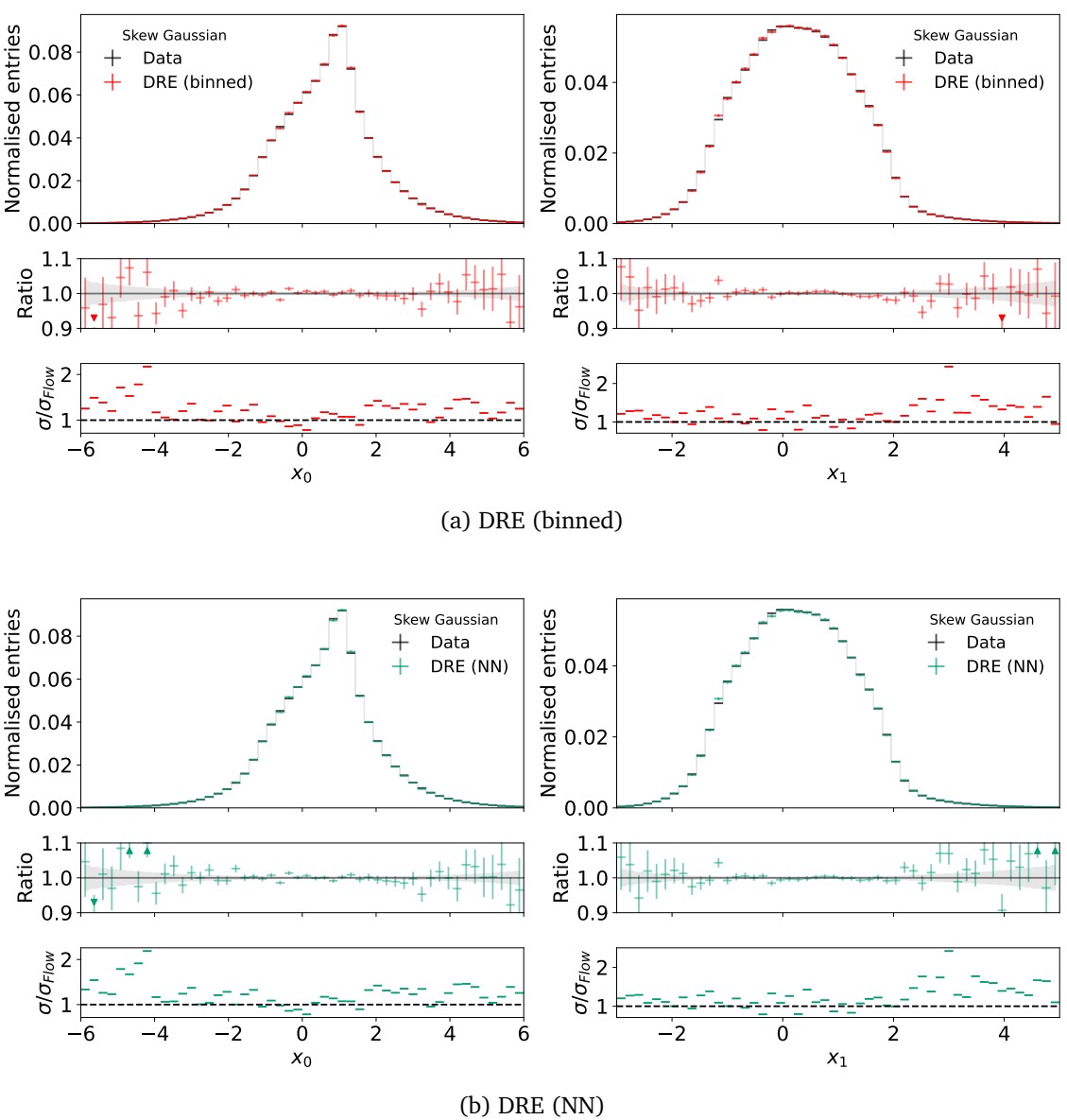

(a) DRE (binned)

(b) DRE (NN)

Figure 3: Closure of corrected conditional distributions to the ground truth when using the reweighting methods with a skewed Gaussian distribution over the conditional quantities. The marginal distributions of $f_0$ and $f_i$ is shown in the upper panels, the ratio to the ground truth distribution in the middle panels, and the ratio between the statistical uncertainty on the reweight methods and that from the conditional normalizing flows in the lower panel. The conditional normalizing flows generally result in strongly reduced statistical uncertainties compared to the reweight methods.

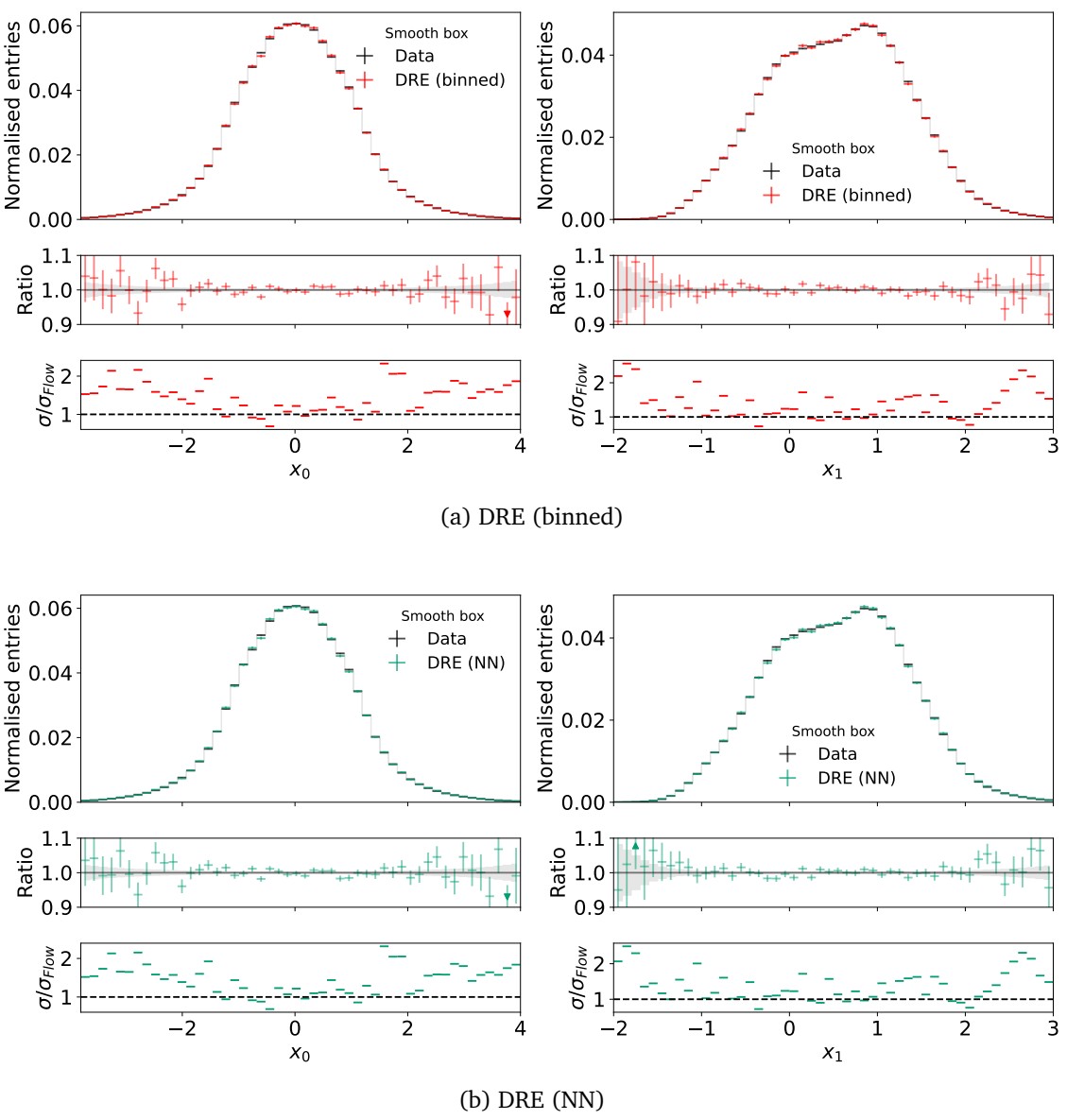

(a) DRE (binned)

(b) DRE (NN)

Figure 4: Closure of corrected conditional distributions to the ground truth when using the reweighting methods with a uniform distribution over the conditional quantities. The marginal distributions of $f_0$ and $f_i$ are shown in the upper panels, the ratio to the ground truth distribution in the middle panels, and the ratio between the statistical uncertainty on the reweight methods and that from the conditional normalizing flows in the lower panel. The conditional normalizing flows generally result in strongly reduced statistical uncertainties compared to the reweight methods.

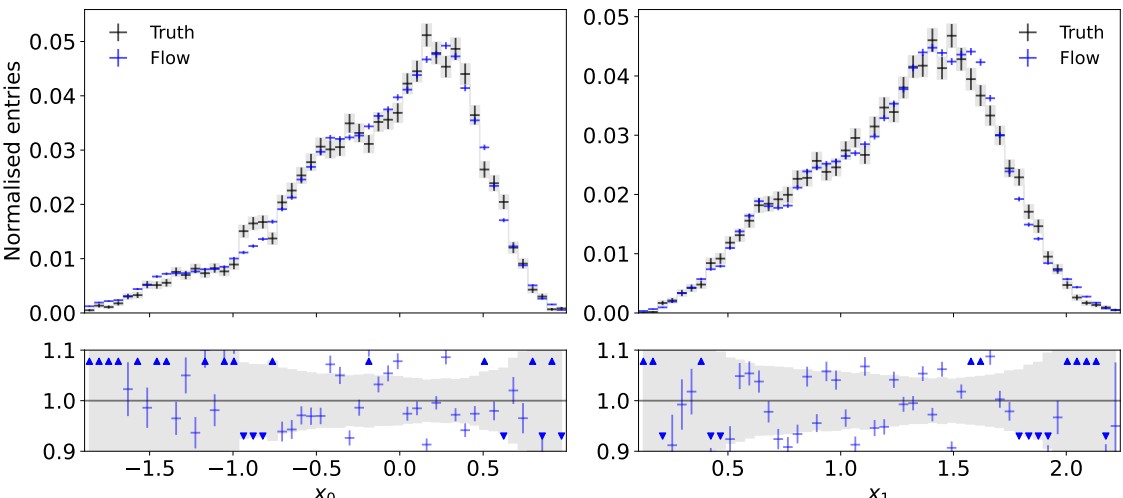

Figure 5: Closure in a small window of conditional values $c_0, c_1 \in (0.9, 1.1]$, for $c_0, c_1 \sim \mathcal{N}(0, 1)$. Samples generated with the normalizing flow (blue) are compared to data drawn from the original distribution (black).

calculated from the observed deviation from the target

$$\text{Pull}_i = \frac{x_i^{\text{pred}} - x_i^{\text{target}}}{\sigma_i}, \tag{7}$$

given the total statistical uncertainty $\sigma_i$ of the predicted and target data. The distribution over all pulls for the distribution should follow a normal distribution. The pull distributions are compared for the three approaches in Fig. 6, where in both cases all three follow a normal distribution. This demonstrates that not only does the cNF result in higher statistical precision, but also that the uncertainties from bootstrapping correctly estimate the true statistical precision. In the case of both reweighting approaches, the statistical uncertainties are derived only from the values of the weights, but the uncertainty on the weights themselves is not included.

Additionally, in practice the target distribution over the conditionals, $q(c)$, is not always analytically known. The limited statistics of target values therefore introduce an additional source of statistical uncertainty into the reweighting approaches. For the cNF this would not change the precision of $p(x|c)$, and the impact on the generated samples for the target distribution would be reduced by sampling multiple times for each $c$ from the base density of the normalizing flow.

One area in which all three approaches lose precision is the case where the target distribution $q(c)$ does not fully overlap with $p(c)$ ($\forall c' \sim q(c)$ where $p(c') = 0$). For these cases the density ratios $R(c)$ are infinite, however as there are no corresponding data in $p(x, c)$ to apply the weights to, the probability density $p'(x, c) = 0$. For the conditional flow $p'(x, c)$ can have non zero probability density for these target values $c$, however this depends strongly on the transformations used and whether it is an interpolation or extrapolation region [29]. In the case of interpolation, many neural network architectures are well behaved and the resulting distributions can be expected to be covered by the statistical uncertainty obtained through bootstrapping. However, for extrapolated values of $c$ the exact behaviour depends strongly on the transformations used in the normalizing flow, and on any regularisation used. It is not expected that the flow will perform well far away from the training region.

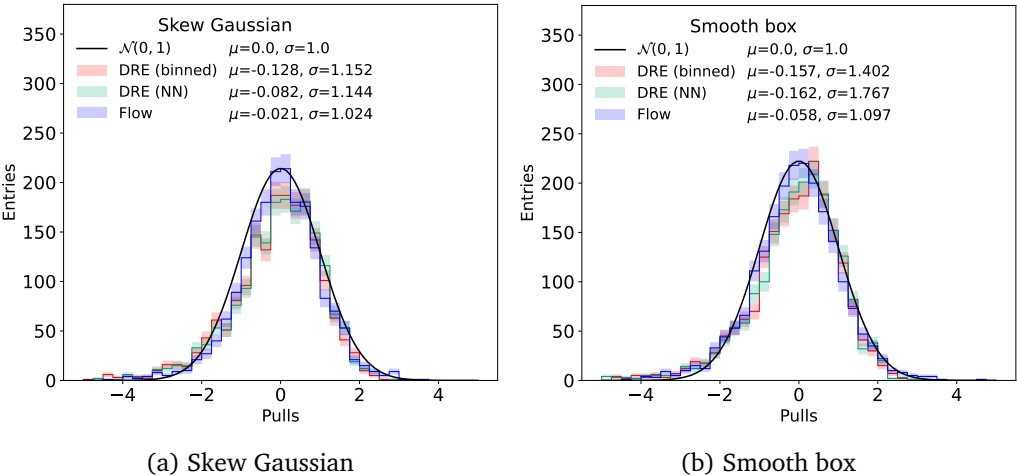

(a) Skew Gaussian

(b) Smooth box

Figure 6: Comparisons are shown of pull distributions over the 2D probability density $p(f_0, f_1)$, approximated using 2D histograms, for the various strategies for correcting distributions over the conditional quantities. The normal distribution is shown for reference in black. Despite resulting in substantially smaller statistical uncertainties on the corrected $p(f_0, f_1)$ prediction, the cNFs maintain appropriate behaviour in the pull distribution, indicating a correct estimate of the statistical uncertainty is obtained via bootstraps.

## 5 Further applications

### 5.1 High energy physics

In many cases, disagreements are observed between the data collected by experiments and predictions; these predictions are often produced via MC sampling generators, such as Herwig [30], Pythia [31], and Sherpa [32]. For example, there is a long-standing discrepancy between various MC generator predictions and LHC collision data in the differential cross sections of pairs of top-quarks as a function of their momenta transverse to the incoming proton beams ($p_T$). Top-quark pair production is ubiquitous in analyses of LHC data due to its relatively high cross-section and striking experimental signature; as such, it has been the subject of intense scrutiny at the LHC. Both the ATLAS and CMS Collaborations have produced detailed measurements of this process and have compared these measurement to state-of-the-art predictions [33–38]. Both experiments have reported differences between data and MC generator simulations. Analyses of the LHC data for which top-quark pair production is a large background often resort to a binned reweighting strategy to account for this discrepancy (see, for example, Refs. [39, 40]). By doing so, they hope to mitigate any mis-modeling of this background process and it impact on their primary analysis target.

In this example we use apply conditional normalizing flow approach to resample $t\bar{t}$ events from simulation according to the measured distributions from data in one kinematic distribution and observe the impact on other measured kinematic distributions. The measured kinematic observables are described in Table 1, and are chosen following a recent measurement of differential $t\bar{t}$ production cross section performed by the ATLAS collaboration [41].

For this study we choose to correct the modelling of the transverse momentum of the top quark $p_T^{t,had}$ to match that in data. However, in principle the method can be used for any combination of the chosen observables, with the impact observed on the remaining distributions.

Top-quark pair production cross sections were predicted with Pythia v8.3 [31] using the Monash tuned set of parameters [42] to simulate $t\bar{t}$ production at the LHC, corresponding

Table 1: Kinematic observables at the particle level for events in top quark pair production in the single lepton channel.

| Observable | Description |
|---|---|
| $p_T^{t,had}$ | Transverse momentum of the hadronically decaying top quark |
| $m^{tt}$ | Invariant mass of the top quark pair |
| $p_T^{tt}$ | Transverse momentum of the top quark pair |

to leading order (LO) + parton shower (PS) accuracy. The NNPDF2.3 QCD+QED LO proton parton distribution set was used [43]. Fiducial requirements were imposed using the Rivet toolkit [44]. The conditional distribution $p(p_T^{tt}, m^{tt}|p_T^{t,had})$ was learned via a normalizing flow conditioned on $p_T^{t,had}$; the cNF was trained using 200 000 $t\bar{t}$ events.

Once $p(p_T^{tt}, m^{tt}|p_T^{t,had})$ is approximated using the Pythia predictions, this is applied to the $p_{\text{data}}(p_T^{t,had})$ observed in data to predict the joint $p(p_T^{tt}, m^{tt})$ distribution by feeding samples of $p_T^{t,had} \sim p_{\text{data}}(p_T^{t,had})$ into the normalizing flow. To achieve this, we approximate the continuous distribution of $p_T^{t,had}$ in data by applying splines to the binned ATLAS measurement. For comparison, we also apply the binned reweighting method using the binned $p_T^{t,had}$ density from Pythia and the unfolded ATLAS data.

Figure 7 compares the outcome of the two approaches to the unfolded data. Neither approach faithfully reproduces the data marginals of $p_{\text{data}}(p_T^{tt}, m^{tt})$, which we attribute to the inadequate predictive power of the leading-order Pythia calculation. Large differences between the resulting distributions are not expected, as in the ideal case both the reweighting and cNF should result in the same corrected probability distribution. Nonetheless, correcting the $p_T^{t,had}$ distribution does yield a better description of the $m^{tt}$ differential cross-sections than the nominal Pythia prediction. Crucially, our approach allows for a more precise model of the corrected Pythia, with 25-50% smaller statistical uncertainties per bin than obtained using the binned reweighting approach.

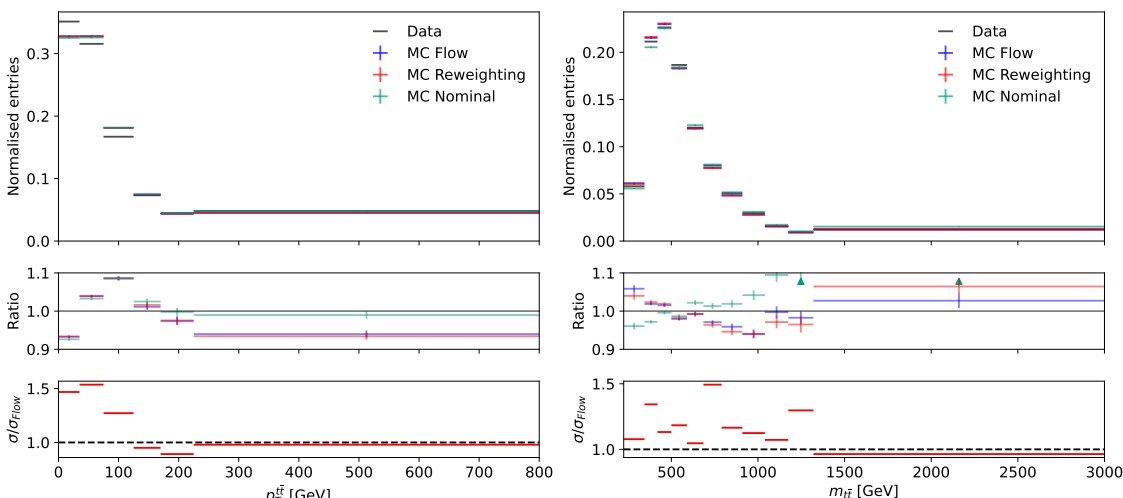

Figure 7: The marginals of the normalizing flow outputs compared to unfolded data. Statistical uncertainties are estimated using bootstraps of the training samples. The cNF approach generally achieves 25-50% smaller statistical uncertainties than the binned reweighting.

## 5.2 Preprocessing for neural networks

Using this approach will be particularly useful for training other neural networks which require distributions over a subset of inputs for different classes to match, for example when using planing to reduce bias. In comparison to reweighting and downsampling approaches, this approach does not reduce the effective training statistics, nor does it introduce a large range of weights per batch. Additionally, it can be used to increase the effective training statistics for a neural network, which has been observed to increase performance in supervised tasks [23].

Furthermore, it can be useful when using distribution or distance based loss functions in conditional generative models which cannot easily incorporate weights. In particular when training partially input convex neural networks [17], the conditional normalizing flow can be used to guarantee the same values of non convex inputs between the input and target batch of data. In cases where there is a difference in the distribution of the non convex inputs between the input and target data, these models can struggle to converge.

## 6 Conclusion

In this work we have introduced an alternative approach to standard reweighting techniques, using conditional normalizing flows. With cNFs we show it is possible to extract a model for the conditional probability distribution and generate new data from the distribution for a desired conditional distribution, removing the need to derive per event weights to modify distributions.

In comparison to binned and density estimation based reweighting techniques, this approach demonstrates better agreement to the data drawn from the true target distribution, and higher statistical precision. Furthermore, the statistical uncertainty on the generated data distributions come only from the number of training examples and can be calculated with bootstrapping approaches, rather than the compound of both the statistical precision on both the training and target distributions. Due to learning the conditional distribution, using cNFs does not suffer from sparse data in high dimensions, like with binned approaches, and instead scales well to higher dimensions. This also enables generating a much larger number of data for given conditional values, which has been seen to be beneficial with generative adversarial networks for training other neural networks.

In addition to the benefits for training other neural networks, we show how this approach can be applied to observed and measured distributions in high energy physics analyses. By training a cNF on the MC simulation of the signal process for the measured observables, mismodelling in one or multiple observables compared to the observed experimental data can be accounted for and the resulting distributions can be compared to the data. This methodology could be further expanded in reinterpretation analyses. The distributions of non observed parameters could be changed, in order to study their impact on the agreement between recorded collision data and the theoretical predictions which depend on these parameters. Examples for such parameters are higher order effects in the signal process, such as the modelling of top quark kinematics or monte carlo tuning parameters. Hypothetical corrections from beyond the standard model physics that modify underlying distributions could also be studied, such as those arising from loop effects in top quark and Higgs boson production mechanisms.

# Acknowledgements

We thank Slava Voloshynovskiy for his helpful discussions and feedback. MA, MG, and JR are supported through funding from the SNSF Sinergia grant called Robust Deep Density Models for High-Energy Particle Physics and Solar Flare Analysis (RODEM) with funding number CRSII5_193716 and the SNSF project grant 200020_212127 called "At the two upgrade frontiers: machine learning and the ITk Pixel detector". MG is further supported with funding acquired through the Feodor Lynen scholarship award. CP acknowledges support through STFC consolidated grant ref ST/W000571/1.

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

# A  Supplementary Material

## A.1  Detailed network architecture

The following table show the hyperparameters of the conditional normalizing flow used in the results section

| Category | Hyperparameter | Value |
|---|---|---|
| | Number of stacks | 12 |
| | Tails | Linear |
| Flow | Number of bins | 10 |
| | Tail Bound | 3.5 |
| | LR scheduler | CosineAnnealingLR(1e-4, 1e-6) |
| | Number of epochs | 500 |
| Training | Training size | 500.000 |
| | Gradient clipping | 10 |

Table 2: Hyperparameters of the conditional normalizing flows used for the toy examples

Table 3: Hyperparameters of the normalizing flows trained on $t\bar{t}$ samples

| Category | Hyperparameter | Value |
|---|---|---|
| | Number of stacks | 5 |
| | Tails | Linear |
| Flow | Number of bins | 10 |
| | Tail Bound | 3.5 |
| | Conditional Base | True |
| | LR scheduler | CosineAnnealingLR(1e-3, 1e-5) |
| | Number of epochs | 500 |
| Training | Training size | 200.000 |
| | Gradient clipping | 10 |

## A.2  Neural network hyperparameters

Throughout the paper we have been using neural networks as a discriminator and for the DRE (NN) method. All the network have the same hyperparameters seen in Table 4.

Table 4: Hyperparameters for the neural networks

| Category | Hyperparameter | Value |
|---|---|---|
| | Hidden layers | 4 |
| | Layer size | 64 |
| Network parameters | Batch norm | After each activation function |
| | Activation function | Leaky ReLu |
| | Loss function | BCE |
| | LR scheduler | CosineAnnealingLR(1e-3, 1e-7) |
| Training | Number of epochs | 200 |

# B Additional plots

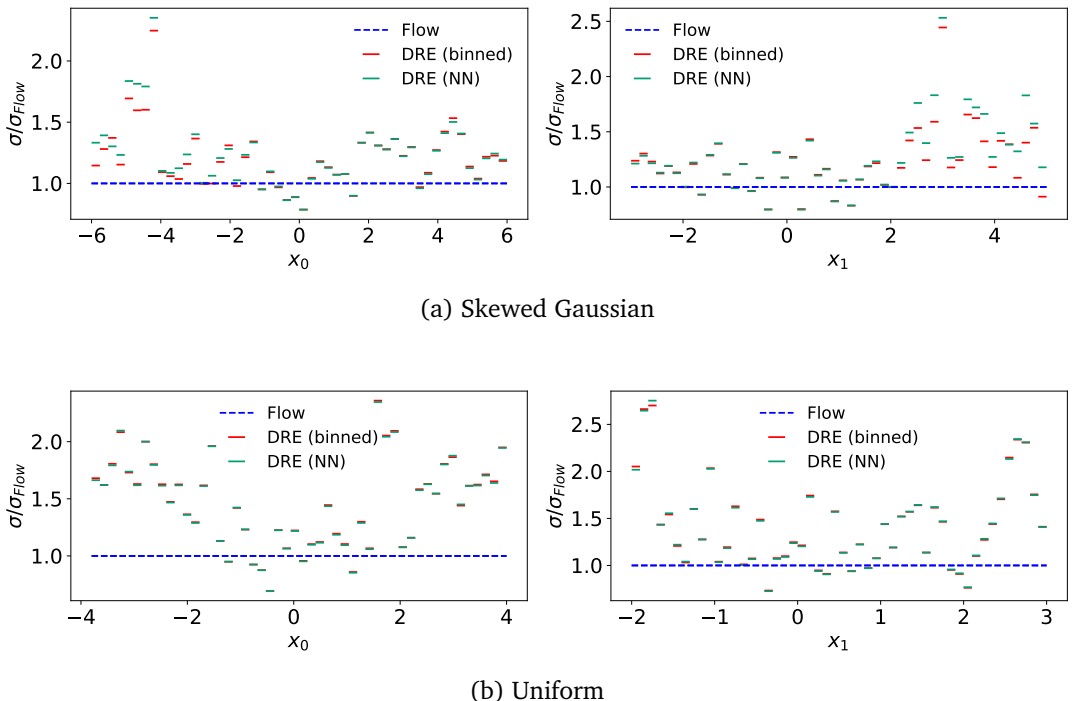

(a) Skewed Gaussian

(b) Uniform

Figure 8: Comparisons are shown of estimated statistical uncertainties on binned marginal distributions over quantities of interest, $f_0$ and $f_1$, after altering the conditional quantities $c_0$ and $c_1$. Both the skewed Gaussian and uniform distribution examples are shown. The conditional normalizing flows generally result in strongly reduced statistical uncertainties compared to the reweight methods.

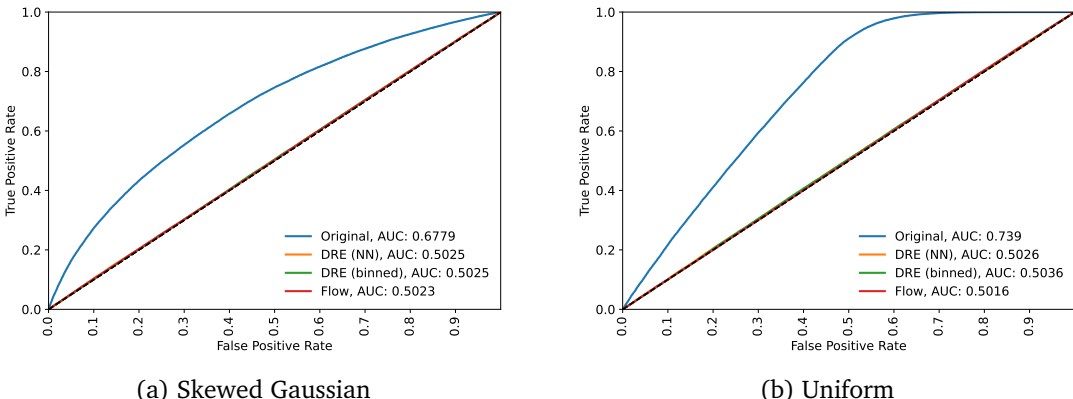

(a) Skewed Gaussian          (b) Uniform

Figure 9: Receiver operator characteristic curves for classifiers trained to discriminate between the ground truth distributions over $f_0$ and $f_1$, with altered distributions over $c_1$ and $c_0$, and those obtained through several strategies for correcting the conditionals' distributions. While a classifier can easily discriminate between the original distribution (produced via a standard multivariate Gaussian over $c_0$ and $c_1$) and the altered distributions, it is unable to do so after correcting the conditional distributions using any of the strategies tested.