# Peer review of "Flow Away your Differences: Conditional Normalizing Flows as an Improvement to Reweighting"

_SciPost Physics_

## Round 1 · Referee Report · Anonymous (Referee 1) · 2024-2-16

Strengths

  1. Robust method for reweighting MC events to data
  2. The model is demonstrated on a real LHC dataset

Weaknesses

  1. Many statements of performance are subjective and not objective
  2. The text is unclear in many places detracting from the ability of the reader to understand what is being accomplished

Report

The authors introduce a novel approach to reweighting Monte Carlo events to agree with experimental data using conditional normalizing flows. They argue that this approach should replace existing data driven methods for obtaining background predictions for searches. While the approach appears promising, it is difficult to determine if the claims of the authors are supported, since many arguments of the improvements are given as subjective statements with no statistical measures to back it up. Therefore, I would recommend that the authors improve objectiveness of their argument for this new approach before being accepted for publication. I gave additional details in how to address these concerns in the requested changes, along with some missing literature and points that could be better clarified.

Requested changes

  1. When introducing the Monte Carlo simulations in HEP, the authors only reference LHC physics. The HEP community is much larger than just the LHC, and this problem applies outside of the LHC as well. To this end, the authors should include a reference to 2203.11110, which discusses the importance of MC simulations in all HEP experiments.
  2. In the introduction, the authors introduce nomenclature of Cover and Thomas, which references a textbook on information theory. Since the typical reader of this article may not be familiar with the nomenclature, the authors should expand upon the definitions to make the nomenclature clear within the text.
  3. In the related work, the authors focus on only applications for reweighting, but miss the related work discussing the other uses of normalizing flows and conditional flows in other areas of HEP, such as for phase space (2001.05486, 2001.05478, 2001.10028, 2011.13445, 2205.01697, 2212.06172, 2311.06198, 2401.09069) or calorimeter showers (2106.05285, 2110.11377, 2312.09290), to point out some of the missing works also using normalizing flows.
  4. In the last sentence before section 4.1, the authors state that the distributions for f(c) for the three scenarios are shown in Fig. 1, but it is unclear from the text if they are referring to the three scenarios for how c is sampled, or the three reweighting approaches. The authors should rework this last paragraph to make it more clear. In general, section 4 to 4.1 could be rewritten to make what their setup more clear to the reader.
  5. In section 4.2, the authors claim that the agreement between all approaches is reasonable. It would be worthwhile if the authors could come up with an objective metric to determine how reasonable the agreement, instead of a subjective statement. And again when they talk about the different regions, it would be worthwhile putting some thought into how to make these statements objective instead of subjective to really emphasize the exact performance gain from this approach.
  6. The authors mention that they train a classifier to separate them from the target distribution, and only mention that they are better than the initial distribution without any support for how much better, which one does the best under this metric, etc. Adding these details along with the ROC curve plot into the main text instead of the supplemental material will be very beneficial to the reader.
  7. In their test for the conditional probability at the end of section 4.2 and figure 5, the authors should supply the chi^2 value or some other metric to measure how close the results are, instead of just stating that they "closely match the original target distribution within statistical uncertainties."
  8. For the test of the pulls, the authors state that the approaches shown in Fig. 6 all follow a normal distribution without any statistical test to validate this. The authors should perform a statistical test for normality such as the Shapiro-Wilk's test.
  9. At the end of section 4.3, the authors discuss regions in which the flow will fail when away from the training data. It might be beyond the scope of this paper, but it would be worthwhile for the authors to consider how one would determine what is "well far away".
  10. Section 5.1 would also benefit from an objective metric determining which one does best. Also, since tools like Sherpa, MadGraph, and PowHEG can simulate this at NLO, is there a reason it was only done at leading order, especially when the authors then claim that the main reason for disagreement is that they only did it at LO?
  11. Overall, the paper could use another read through to improve the clarity of the text and more explicitly explain what is being done so that a non-expert could understand.

---

## Round 1 · Referee Report · Prasanth Shyamsundar (Referee 3) · 2024-3-13

Strengths

  1. A novel approach for morphing/enhancing simulations using measurements.
  2. The paper is overall well-written.
  3. The technique shows promising results.

Weaknesses

  1. I have some concerns regarding the uncertainty estimation.
  2. Presentation could be improved in some places.
  3. Some useful details and discussions are missing in the paper, in my opinion.

Report

This work proposes the use of conditional generative models (normalizing flows in particular) for morphing simulated data, by forcing the distribution of some event attribute $c$ to match a target distribution (derived possibly from measurements). This is an alternative to reweighting based approaches for the same task. The idea is fairly novel, the paper is overall well-written, and the authors demonstrate a substantial improvement in precision when compared to the reweighting based approaches.

I have some concerns and recommendations, many of them regarding the estimation of uncertainties. I would recommend publication of the paper in this journal with some minor changes.

Requested changes

Thoughts/concerns/recommendations regarding uncertainty quantification:

Out of the following list, I strongly recommend that points 4 and 5 are addressed. I'm okay with the authors ignoring 1, 2, 6, and 7. 3 has no recommendations attached to it and just expands on 1 and 2.

  1. In the toy example, there are three sources of randomness in the procedure used to produce the $x$-histograms using the cNF approach. The first is the randomness in the training of the neural network (including the randomness in the training data). The second is the randomness in the $c$ values fed into the network. The third is the randomness in the $z$ values that get mapped to $x$ values, for a given trained network and a given set of $c$ values. In the bootstrap approach for estimating errors, as described in the paper, the neural networks are trained multiple times using bootstrapped training data. This covers the first source of randomness. I'm assuming that in each bootstrap, different $z$ values are used to produce $x$-s, which covers the third source of randomness. However, the second source of randomness is not covered by this process, since the exact same $10^7$ datapoints are used in each bootstrap. I believe this would lead to an overall underestimation of the errors. Recommendation: The $c$ values (from the target distribution) fed into the trained network should also be bootstrapped, i.e., for each trained network, a different bootstrap of $c$-values should be used to produce the $x$ histograms.

  2. In equation 7, the $x^\mathrm{target}_i$ used in the definition of the pull is the observed normalized $x$-histogram count, computed using the data from the target distribution. In my opinion, $x^\mathrm{target}_i$ should be the true expectation under the target distribution (in the infinite sample limit). In a typical histogram-based analysis, the error bar on a background estimate is supposed to indicate the typical size of the deviation from the true expectation under the background. If this true (infinite sample limit) background expectation is not known, another option is to sample a separate set of, say 10^7, events from the target distribution and estimate $x^{obs}_i$ and statistical error $\sigma^{obs}_i$ using this dataset. Now a 2-sample pull can be defined as $\frac{x^\mathrm{pred}_i - x^\mathrm{obs}_i}{~\sqrt{(\sigma^\mathrm{pred}_i)^2+(\sigma^\mathrm{obs}_i)^2}~}$.

  3. My guess is that the effects of a) not bootstrapping the $c$-values and b) using $x^{target}_i$ from the exact same dataset that provides the $c$-values fed into the trained flow model are cancelling each other to give a good distribution of pulls. An argument can possibly be made for why the current uncertainty estimation and pull calculation procedures are fine as they are. In my opinion, it all depends on how the morphed $x$-histogram estimates and the associated errors will be used in the subsequent analysis. Things are sufficiently complicated here (with the morphed simulation-based-predictions being correlated with observed data) that I'm actually okay if the authors choose to ignore the two points above.

  4. With only a small number (12) of bootstraps being used, there will be large errors in the estimated uncertainties (standard deviation of the error-estimates will be around $0.2\times \mathrm{actual~errors}$). This needs to be acknowledged in the paper even if the number of bootstraps cannot be increased easily. This mis-estimation of the error bars will not be caught by the closure test using pulls because a) the errors on the error-estimates will be positive sometimes and negative at other times and b) the errors on the error-estimates will be roughly uncorrelated with $x^\mathrm{pred}_i-x^\mathrm{truth}_i$.

  5. The exact formulas used in the bootstrap-based error-estimation could be provided, especially considering that only 12 bootstraps are used. Computing the standard deviation among the bootstraps with degrees of freedom set to 12 vs 11 amounts to a ~5% effect, so it would help to be specific.

  6. It is troubling that in Fig 6, the apparently standard DRE technique fails the closure test, even for a toy example where $q(x|c) \equiv p(x|c)$, which is the best case scenario for such techniques. This is briefly explained in page 10 as: "In the case of both reweighting approaches, the statistical uncertainties are derived only from the values of the weights, but the uncertainty on the weights themselves is not included." However, considering the complexity of the situation (using observed data in some variable to morph the simulated predictions, which will be used as background estimates in a subsequent analysis in a different variable), I would be skeptical of the estimated errors unless the closure test is passed in some simple examples, at least. I would recommend that the authors attempt the following simpler toy example where $c$ and $x$ are categorical variables taking, say, 10 discrete values each. $p(c)$ can be parameterized with 1 probability vector of length 10 and $p(x|c)$ can be parameterized with 10 conditional probability vectors of length 10. Now with a finite dataset, both techniques (reweighting and learning conditional probabilities) can be performed using histograms instead of neural networks. Satisfying the closure test for both methods simultenously in this example would give some confidence that the estimated-errors have the same meanings under both approaches.

  7. With the techniques considered in this paper, the uncertainties in the different histogram-bins are highly likely to be correlated. In addition to estimating the variances of the bin-wise predictions with bootstraps, it might be interesting to look at the covariance matrix as well (though 12 boostraps may not be sufficient to study this).

Other changes requested:

  1. It is unlikely that the cNF-based approach will be universally better than DRE in all situations. As a simple counter example, let's say $q(c)$ and $p(c)$ are almost the same, so that the weights are very close to 1. Furthermore, let's say $p(x|c)$ is very difficult to estimate. Now the DRE approach will perform really well, because there's basically no morphing required. On the other hand the cNF approach might struggle (especially if the amount of training data is limited). In general the performance of the techniques will depend, in non-trivial ways, on $p(c)$, $q(c)$, $p(c|x)$ and the amount of training data available. This could be mentioned/discussed in the paper.

  2. The description of the toy example in page 5 is very difficult to follow. $c_0$ and $c_1$ are real valued parameters but $c_0'$ and $c_1'$ are distributions, which is confusing. $f_0(c_0', c_1')$ is not the distribution of $(c_0', c_1')$ but the distribution of $x_0$, which is confusing. In equation 5, it is unclear what $c_0', c_1' =$ (a 2d-Gaussian) means. Do $c_0'$ and $c_1'$ correspond to the marginals of the 2d-Gaussian? If so, then the off-diagonal terms of the covariance matrix don't matter, so I'm not sure. In the description of skewed Gaussian and symmetric smooth box distributions, $c_1$ and $c_2$ should be $c_0$ and $c_1$, I think. I recommend describing the toy example in this format (even if these equations may not be correct):

    $$p(x_0, x_1 | c_0, c_1) = p(x_0 | c_0,c_1)~p(x_1 | c_0,c_1)$$
    where
    $$p(x_0 | c_0, c_1) = \frac{g_0(x_0 ; c_0,c_1)}{1+g_1^2(x_0 ; c_0,c_1)} - \frac{g_1(x_0 ; c_0,c_1)}{1+g_0^2(x_0 ; c_0, c_1)}$$
    $$g_0(x_0 ; c_0,c_1) = \mathrm{normalpdf}(x_0 ; \mu=c_0, \sigma^2=2c_0)$$
    and similarly for $g_1$, and similarly for $p(x_1 | c_0, c_1)$.

  3. In the toy example, $f_i(x_i | c_0, c_1)$ don't appear to be standard distributions. Can you explain how you sampled datapoints from these distributions? Additionlly, can you provide a motivation for this choice of distributions?

  4. With the discussion and equations in page 1, it is explained how altering the marginal distribution from $p(c)$ to $q(c)$ will improve the multidimensional distribution of $(c,x)$. While it is technically true that $D(p'(c,x) \| q(c,x)) <= D(p(c,x) \| q(c,x))$, the argument is misleading in my opinion. Under this procedure, the simulated distribution $p(c)$ has been manually forced to fit the observed distribution $q(c)$, so the distribution of $c$ cannot be used for inference purposes. A more important question is whether $p'(x)$, the marginal distribution of $x$ under $p'$, will be better than $p(x)$, since $x$ is the variable one can perform inference with. I don't think it is guaranteed that $D(p'(x) || q(x)) \leq D(p(x) || q(x))$. This caveat could be included in the introduction.

  5. More details about the physics example tackled in the paper could be provided. For example, the Pythia and experimental distributions of $p_\mathrm{T}^{t,had}$ could be provided, so the reader can see the extent to which they differ.

  6. The meaning of the tiny triangles, for example in figures 4 and 5, was not obvious to me. It could be included in the paper.

---

## Editorial Decision

awaiting_resubmission